# Masking Effect of *Cassia grandis* Sensory Defect with Flavoured Stuffed Olives

**DOI:** 10.3390/foods11152305

**Published:** 2022-08-02

**Authors:** Ismael Montero-Fernández, Jhunior Abrahan Marcía-Fuentes, Gema Cascos, Selvin Antonio Saravia-Maldonado, Jesús Lozano, Daniel Martín-Vertedor

**Affiliations:** 1Department of Agricultural and Forestry Engineering, School of Agrarian Engineering, Universidad de Extremadura, 06007 Badajoz, Spain; 2Faculty of Technological Sciences, Universidad Nacional de Agricultura, Catacamas 16201, Honduras; jmarcia@unag.edu.hn; 3Technological Institute of Food and Agriculture CICYTEX-INTAEX, Junta of Extremadura, Avda. Adolfo Suárez s/n, 06007 Badajoz, Spain; gcascosc01@educarex.es; 4Faculty of Earth Sciences and Conservation, Universidad Nacional de Agricultura, Catacamas 16201, Honduras; saraviaselvin@yahoo.com; 5Doctoral Program in Sustainable Territorial Development, Science Faculty, Universidad de Extremadura, 06007 Badajoz, Spain; 6Perception and Intelligent Systems Research Group, Universidad de Extremadura, 06006 Badajoz, Spain; jesuslozano@unex.es; 7Research Institute of Agricultural Resources INURA, Avda de la Investigación s/n, Campus Universitario, 06006 Badajoz, Spain

**Keywords:** *Cassia grandis*, sensory analysis, volatile composition, E-nose, odour, table olives

## Abstract

Carao (*Cassia grandis*) is an America native plant characterized by its high iron content. This particular property allows its use as a natural additive to fix the black colour in California-style black olives, while masking its unpleasant aroma by stuffing olives with flavoured hydrocolloid. The tasting panel evaluated olives filled with unflavoured hydrocolloid with a fruity aroma, classified them as an extra category. Olives with the Carao addition presented a positive aroma, but also showed negative sensory attributes such as cheese, fermented and metallic flavours/aromas. The aroma of lyophilized Carao was better than the fresh one. The ‘Mojo picón’ aroma masked defective olives, allowing their classification from the second to the first commercial category. The volatile compounds belonged to the following families: terpenes, hydrocarbons, and oxygenated compounds, while the minor ones were alcohols and acid derivatives. The main volatile compounds identified were dialyl disulphide and 3-methyl-butanoic acid; among the minor ones were 2,4-dimethyl-hexane and dimethyl-silanediol and nonanal. Addition of fresh Carao increased the unpleasant aroma provoked by 3-methyl-butanoic acid, 2-methyl-butanoic acid and (E)-2-Decenal. Finally, an electronic device was able to discriminate these aromas and the results obtained agreed with those of the tasting panel and the volatile compounds.

## 1. Introduction

The Carao (*Cassia grandis*) is a tree of the *Fabaceae* family native to Central America, the Caribbean, and the northern region of South America, found at about 900 m altitude on open roads, pastures, and roadsides [1]. The fruit is a woody legume that reaches up to 50 cm long and more than 5 cm thick and, when it ripens, is black with transverse seeds [2]. Its use in alternative medicine has become one of the key elements in pharmacology, as it is rich in metabolites such as steroids, terpenes, essential oils, reducing sugars, amino acids, amines, saponins, glycosides and polysaccharides. This plant is rich in elements that have anti-anaemic properties such as potassium, magnesium, cobalt, nickel, and iron [3]. Authors [4,5] have shown that Carao in vivo has anti-anaemic potential properties due to its high iron content and high bioavailability. In South America, Carao is also used as a strategy for use in nutritionally modified foods, to improve the bioavailability of nutrients, as is the case of the egg, which can be fortified with Carao to combat the anaemia due to its high iron content [3].

The addition of Carao during the preparation of table olives could be an alternative for Californian-style oxidized black olives because an iron salt, such as ferrous gluconate is added to fix the colour of the olives. This style is one of the most common preparation processes for table olives in the world [6,7]. The preparation begins by preserving olives for at least 4 months in salt and acid in which oxidation of the olives begins. Next, several lye treatments with NaOH are carried out to eliminate their bitterness, while simultaneously applying forced air to oxidize the olives, which changes their colour from green/brown to black. The colour of the olives is finally fixed by the addition of an iron salt. The black olives produced are put in cans with a brine solution and because the final product obtained is not microbiologically stable, it must be subjected to a thermal sterilization process at 121–126 °C for 25–45 min [8]. The nutritional properties of table olives are mainly influenced by the processing method used and by pre-harvest factors [9].

The high temperature provokes the degradation of phenols, vitamins and other chemical compounds and stimulates the oxidative degradation of the olives, leading to the generation of non-characteristic aromas and flavours (off flavours) in the final product [10,11,12]. Therefore, it is common to find studies [12,13,14] in which natural plants are added into the cans to improve the aroma and increase the bioactive compounds content in table olives.

Conventional analytical techniques such as gas chromatography and a specialized tasting panel are powerful tools to determine the aromatic characteristics of the olives obtained [12,14,15]. These techniques can discriminate olives based on positive and negative attributes and allow them to be classified according to International Olive Council [16,17]. However, for industries, these types of analysis are expensive and require a long analysis time [18].

The electronic nose (E-nose) is a device including a combination of sensors that allows samples with different aroma profiles to be discriminated in different types of matrices: wine [19], edible oils [20], citrus alterations, fungal infections [21,22], or discrimination of the quality and geographical origin of olive oil [22]. In this sense, the E-nose, due to its speed and low-cost, can be an alternative to chromatography and sensory analysis by a tasting panel, for assessing the presence of defects. This is an important aspect for the subsequent classification of the product according to its flavour profile. Therefore, the electronic device can be used at an industrial level as a fast, powerful, and effective tool to evaluate the volatile organic compounds of table olives, including the cooking defects [7,23] or abnormal fermentation [24,25]. A limitation of the use of the E-nose is that it does not allow the quantification of the volatile compounds in the aroma of the food and auxiliary techniques are needed to identify them such as a tasting panel or GC-MS [25]. For all of that, the main objective of this work was to study the effect of masking the aroma caused by the addition of *Cassia grandis* in Californian-style black olives, by stuffing olives with hydrocolloids flavoured with ‘Mojo picón’ at different concentrations to produce olives with a high level of iron in a natural way.

## 2. Materials and Methods

### 2.1. Samples

The experimental design is illustrated in Figure 1. 

Olives (*Olea europaea* L.) of the ‘Manzanilla Cacereña’ variety at a green stage of maturity were harvested during the 2021/22 crop season in a collaborative research at the CICYTEX Research Centre of Extremadura (Spain). Olives were stored in tanks with acetic acid solution (3% *v*/*v*) under aerobic conditions for 4 months. 

Carao fruits (*Cassia grandis*) were collected in the city of Choluteca, department of the same name (Honduras) A total of 500 g of fruit from different parts of the central area of the tree was collected, with three repetitions. After harvested, the fruit were transported to the Biotechnology Laboratory of the National University of Agriculture, Catacamas, Olancho 16201, Honduras.

### 2.2. Experimental Design 

#### 2.2.1. Carao Fruit Preparation

Carao fruit was manually peeled. One portion was stored at frozen conditions (−80 °C) until they were lyophilized (LIOTOP model L 101). Subsequently, the lyophilized plant was ground with a commercial mill (LABOR model SP31) to get a particle size of 0.5 to 0.3 mm [26]. The fresh plant was vacuum-packed in plastic bags and they were stored at frozen conditions (−80 °C) until they were used for the study.

#### 2.2.2. Californian-Style Black Olive Preparation Process

Next, olives were removed from storage conditions, washed, and processed according to the Californian-style oxidized black olive method [12]. They were introduced into an oxidized tank with air bubbled continuously to oxidize them, and meanwhile, olives were treated with NaOH to remove the bitterness. After neutralization with lactic acid and CO_2_, different additives were added to fix the olives colour: (i) ferrous gluconate solution (0.015%, *w*/*v*); (ii) fresh *C. grandis* at 1:5 ratio; and (iii) lyophilized *C. grandis* at 1:5 ratio. The experiment was done in quintuplicate.

#### 2.2.3. Stuffed Olives with Flavoured Hydrocolloids

Olives were pitted for the subsequent stuffing. The different hydrocolloid fillings were prepared with a water base to which was added a 2:1 ratio of sodium alginate and guar gum and 0%, 2% and 4% ‘Mojo picón’ commercial flavouring (Neroliane, Grasse, France) [23,27]. Three groups of samples were made with table olives in which colour was fixed with ferrous gluconate, fresh and lyophilized *C. grandis*: (i) olives stuffed with 0% of ‘Mojo picón’ flavour (T1, T2 and T3); (ii) olives stuffed with 2% of ‘Mojo picón’ flavour (T4 and T6); and (iii) olives stuffed with 4% of ‘Mojo picón’ flavour (T5 and T7). 

Olives were filled manually with a syringe. The olives were then soaked for 24 h in a 0.25% CaCl_2_ solution, washed and introduced into jars (150 g) with a brine that contained 2% of NaCl and 0.015% of the additive. Finally, the jars were sterilized in an autoclave at 121 °C for 30 min. Sensory analysis, analysis of volatile compounds and the E-nose measurement were made for all samples.

### 2.3. Analyses

#### 2.3.1. Sensory Analysis

A sensory analysis was done by eight expert panellists from the research centre CICYTEX (Extremadura, Spain) and the University of Extremadura. They were formed following the recommendation of International Olive Council [16]. For the sensory analysis evaluation of the stuffed olives, a score board was prepared with a structured scale from 1 to 11 points based on perceived positive and negative odours. In addition, the panellists were asked to describe other sensory attributes of the samples studied. Three olives were placed in standard glasses with 10 mL of brine. The trained panel rated the intensity of all the aromas perceived. Sensory analysis results were considered valid when the coefficient of variation was less than 20%.

#### 2.3.2. Analysis of Volatile Compounds

Table olives were analysed by gas chromatography for volatile organic compounds (VOCs) determination according to Sánchez et al. [27]. For the analysis, 2 g of sample were crushed and homogenized in a glass vial with 7 mL of NaCl (30% *w*/*v*). The aromas of the samples were absorbed with a polydimethylsilane/divinylbenzene (PDMS/DVD) Stable Flex fibre (65 μm Supelco at 40 °C for 30 min. Samples were injected into Agilent DB WAXetr GC-MS model 456-CG triple quadrupole gas chromatograph with a capillary column (60 m × 0.25 mm × 0.25 mm) from Agilent Technologies (Palo Alto, CA, USA). The identification of the VOCs was done on mass spectra matching with the standard NIST 2.0 MS library.

#### 2.3.3. E-Nose Measurements

The E-nose was designed by the Department of Electrical, Electronic and Automatic Engineering of the University of Extremadura. The device consisted of 11 metal oxide semiconductor (MOX) sensors, which were specifically described in Sánchez et al. [27]. The E-nose included analogue and digital electronics combined with the sensing elements on a single chip installed in a developed micro-board (Table 1). It contained a +3.7 V lithium battery and the device could communicate via Bluetooth with a mobile phone application.

The samples measured with the E-nose were analysed under the same conditions as the sensory analysis [16]. The analysis using the E-nose took place in two phases: an adsorption phase, in which the sensors were placed in contact with the headspace of the samples for 60 s, and a 30 s desorption phase, in which the sensors were only placed in contact with air, which serves as a reference signal.

### 2.4. Multivariate Data Analysis 

The measurements obtained with the E-nose were analysed using Principal Component Analysis (PCA) [28] as an exploratory data analysis. Next, Partial Least Squares Discriminant Analysis (PLS-DA) [29] was performed to identify the components or latent variables (LV) discriminating the most between the different sample groups. A confusion matrix was constructed to deduce cross-validation predictions. From this matrix, the percentage of correct predictions of the sum of the diagonal elements found was calculated. Data analysis was performed using the software Matlab version R2016b, version 9.1 (The Mathworks Inc., Natick, MA, USA) with PLS_Toolbox 8.2.1 (Eigenvector Research Inc., Wenatchee, WA, USA).

### 2.5. Statistical Analysis

Significant differences and homogeneous groups of means were established by analysis of variance (ANOVA). When the difference between mean values was significant, a test of comparison of the means was performed using the Tukey method (univariate analysis). Mean values and standard deviation are reported. The software IBM SPSS version 19 (IBM® SPSS®, Spain) for Windows was used in the statistical treatment of the data.

## 3. Results and Discussion

### 3.1. Sensory Analysis of Spanish-Style Table Olives

Californian-style table olives were sensorially evaluated by the tasting panel (Table 2) following the methodology stablished by the International Olive Council [16]. 

Olives stuffed with non-flavoured hydrocolloids presented a fruity aroma with a slight cooked odour caused by the thermal sterilization treatment [7]. They were classified as the highest commercial category: extra. The fixing of the black colour of these olives was carried out in the way that the company usually does, using ferrous gluconate. On the other hand, the colour of another batch of olives was fixed with fresh *C. grandis*, an exotic plant rich in iron. These olives presented a positive aroma related to a fruity and sweet odour. The cooking defect of the olives disappeared when Carao and/or different concentrations of ‘Mojo picón’ aroma were added. However, they showed negative sensory attributes, such as a cheesy, fermented, and metallic flavours/odours, caused by the addition of *C. grandis* during processing. The intensity of the odour defect was high, being classified in a worse commercial category: the second category (4.5 < PPD ≤ 7.0). Therefore, these olives were filled with a flavoured hydrocolloid with ‘Mojo picón’ to try to mask the possible defects in the olives caused by the addition of the plant. Then, the panellist decreased the intensity of the defect perceived due to the positive odour of the aroma added to the stuffed olive. The ‘Mojo picón’ aroma at 2% resulted in olives with less intensity of the defect, which were then classified in a better commercial category, the first category (3 < PPD ≤ 4.5). However, when olives were filled with flavoured hydrocolloid at 4%, the defect disappeared, being classified as an extra category. Therefore, it is interesting to note that the sensory defect caused by the addition of *C. grandis* plant for black colour fixation was balanced by filling olives with ‘Mojo picón’ aroma that resulted in positive aromas predominating over the negative ones. Sánchez et al. [23] evaluated the olfactory pattern of Californian-style black olive stuffed with flavoured hydrocolloids to mask the cooking defect provoked by thermal sterilization treatments. 

On the other hand, the black colour of the olives was fixed by *C. grandis* subjected to a freeze-drying process, during the Californian-style preparation process. In this case, olives presented positive aromas with a high fruity intensity, with a sweet smell and a slight touch of toast, significantly higher than when the colour was fixed with fresh *C. grandis*. Note, that the use of this plant as a lyophilized food additive also involved the appearance of slight sensory defects of cheese and fermented flavours. These defects were practically negligible and less than when the fresh plant was added. In fact, these olives were classified as the extra category. However, an attempt was also made to eliminate the slight sensory defects by filling the olives with ‘Mojo picón’ flavoured hydrocolloid. In this case, with the addition of the aroma at 2%, the perception of the fruity and sweet aroma of the olives decreased, increasing the characteristic aroma of ‘Mojo picón’. The sensory defects were not perceived by the tasters and olives were again commercially classified as the extra category. The same occurred when the aroma concentration was increased to 4%. No sensory defects were found by the tasters, and olives were again classified in the highest commercial category. Legally, olives can be stuffed with different types of aromas that can serve as a marketing strategy to mask sensory defects with different intensity, so long as the olives maintain their physical characteristics intact.

### 3.2. Volatile Compounds of Californian-Style Table Olives

The range of volatile organic compounds (VOCs) was classified into the following families: derivatives of acids, alcohols, carboxylic acids, hydrocarbons, oxygenated compounds, and terpenes. The results for the different treatments are shown in Figure 2. 

Significant differences were observed between the VOCs analysed in the different experimental treatments. The families of VOCs that were found in greater proportions were terpenes, hydrocarbons, and oxygenated compounds, while the minor ones were alcohols and acid derivatives. The acid derivatives increased considerably in T2 compared to the rest of the treatments (49.89%), due to the contribution to VOCs by the Carao pulp to the olives. In the remaining treatments, the concentration of acid derivatives decreased, 27.9% for T3 and 44.9% for T5, while the reduction for T6 was 90.9% and 94.6% for T7. The acid derivatives of the T4 treatment were not identified. The next group of VOCs was the alcohols, the majority in T1 and with a lower concentration for the rest of the treatments, even being undetected in T6. The highest concentration of carboxylic acids was observed in T1. The rest of the treatments decreased considerably, not being detected even in the T4 treatment. The treatments that presented a lower decrease in carboxylic acids with respect to T1 were T5 (27. 8%) and T6 (69.8%). The next group of VOCs was the hydrocarbons. The T1 presented a low concentration of this type of compound (4.4%), increasing its concentrations in the different treatments. The treatments that had a higher concentration in this family of VOCs were T4 (85.2%), T6 (85.4%) and T7 (61.7%). The oxygenated compounds (aldehydes and ketones) were present in all the experimental treatments. The highest concentration of these compounds was in T2 (20.1%), followed by T5 (9.5%) and T1 (26.7%). The lowest concentrations were observed at T6 and T7. The last group of VOCs identified was the terpenoids. These constituents were not detected in T1 and were found in low concentrations in T2. The highest concentrations of this family of compounds were obtained in T4 (51.2%), T6 (45.6%) and T7 (51.2%).

By adding fresh Carao to the olives, there was an increase in the concentration of carboxylic acids, hydrocarbons and oxygenated compounds with respect to T1. However, carboxylic acids and alcohols decrease. When ‘Mojo picón’ was added to the olives to mask the smell of Carao, it was observed that at a concentration of 2%, the terpenoids increased considerably, decreasing the concentration of hydrocarbons, oxygenated compounds and alcohols. On the other hand, when ‘Mojo picón’ was added at a concentration of 4%, the acid derivatives were present in low concentration and there was an increase in the carboxylic acids. However, the terpenoids decreased. On the other hand, when lyophilized Carao was added to the olives, there was an increase in terpenes and hydrocarbons with respect to T1 and fresh Carao. On the other hand, there was a decrease in oxygenated compounds, acid derivatives and carboxylic acids after the addition of lyophilized Carao. By adding ‘Mojo picón’ at a concentration of 2% to the olives with fresh Carao to mask the smell of Carao, the concentration of terpenes and carboxylic acids increased compared to fresh Carao. Instead, there was a decrease in acid derivatives and oxygenated compounds. For the addition of ‘Mojo picón’ in a concentration of 4%, the terpenoids increased with respect to the concentration of 2% and the presence of alcohols in low concentration was also observed.

The profile of VOCs for the different treatments applied to Californian black olives is shown in Table 3. In total, 36 VOCs were identified, which were grouped into different families of volatile organic compounds. The main of VOCs were dialyl disulphide and 3-methyl-butanoic acid, among the minor ones are 2,4-dimethyl-hexane and dimethyl-silanediol and nonanal.

The principal carboxylic acid derivative found was 3-methyl-butanoic acid, which was only present in T2 (17.7%) and T3 (8.1%). The presence of VOCs is due to the addition of fresh and lyophilized Carao to the oxidized black olives. Its concentration was higher in fresh than in lyophilized Carao. This compound provides an unpleasant smell of cheese, faecal, putrid fruit, rancid and sweat [30]. These results agree with those detected by the panel of tasters. Another of the acid derivatives was the isomer 2-methyl-butanoic acid, also present in T2 (8.8%) and T3 (3.8%). Another carboxylic acid derivative that was highlighted was ethyl cyclohexanecarboxylate, whose concentration was significant in T1 (16.5%) and insignificant in T2 (5.6%) and T5 (6.4%). For the rest of the treatments, this compound was not detected. Additionally, the methyl-ester-2-propenoic acid (cinnamic acid) was notable in T5 (2.7%), T6 (1.5%) and T7 (0.9%) and responsible for the fruity smell [31], whose properties include its antioxidant and antimicrobial characteristics [32]. This compound results from the addition of ‘Mojo picón’ aroma to the olive filling to mask the smell coming from Carao addition.

On the other hand, within the family of alcohols, two compounds were detected; dimethyl silanediol, which was absent in T1, so its presence may be due to the addition of Carao. This compound provides a body odour [33]. The other alcohol detected was 2-methoxy-4-methylphenol, whose highest concentration was found in T1 (20%) and was maintained in some of the treatments. This compound provides a positive smell of spices and cloves [34]. Among the carboxylic acids, creosol from olives was found. Its main concentration was in T1 (25%), decreased for the remaining treatments, and was not detected for T4 and T7. This compound provides a woody smell [35]. The other carboxylic acid present was benzoic acid, whose main concentration was found for T5 (19.2%) followed by T6 (7.3%) andT1 (7.4%). Its compounds present an unpleasant urine odour [36].

A total of 10 compounds of the hydrocarbon family were identified. The most prominent were diallyl disulphide in T3 (12.8%), T4 (13.2%), T6 (18.8%) and T7 (17.1%). This compound must come from the addition of ‘Mojo picón’ in flavoured fillings of table olives. This constituent is found mainly in garlic and has antimicrobial properties [37]. Another important hydrocarbon determined was 13,7-dimethyl-6-octadien-3-ol, being found in highest concentrations for samples T2 (8.7%) and T3 (4.5%), being higher for fresh Carao. This compound must come from the addition of Carao.

Another group of compounds was the oxygenated compounds, formed by aldehydes and ketones. Among them, benzaldehyde stands out, whose concentration was higher for T1 (23.6%) and decreased for the rest of the treatments, not being detected in T5, T6 and T7. This compound provides a positive odour of almonds [38] and fruity odours [39]. Within this group of compounds, (E)-2-decenal was also the most common, whose aroma comes not only from the olive without the addition of Carao (T1) but also from Carao and ‘Mojo picón’. The highest concentration was found for T5 (12.5%). The last family of compounds found were terpenes or isoprenoids [40], which are secondary metabolites found in the essential oils of plants and have antimicrobial and antioxidant actions. Among them, p-cymene acid stands out, whose highest concentrations were found in T4 (15.1%), T6 (13.8%) and T7 (15.6%). This aroma comes mainly from the ‘Mojo picón’ added to the olive fillings. This terpenoid is a precursor to other compounds such as thymol and carvacrol [41].

With the results obtained, it was observed that the addition of fresh Carao causes an increase in VOCs with an unpleasant odour, such as 3-methyl-butanoic acid and 2-methyl-butanoic acid and (E)-2-decenal. When adding lyophilized Carao, these compounds were the ones that also had an unpleasant odour but one in lower concentrations than in fresh Carao. However, it seems that the addition of aroma in the olive fillings caused a dilution effect of the unpleasant aroma, as the VOCs responsible for the ‘Mojo picón’ odour increased, such as: diallyl disulphide, di-2-propenyl-trisulphide, allyl trisulfide and β-Terpinene.

### 3.3. E-Nose Application to Stuffed Olives with Flavoured Hydrocolloid

To verify the differences in VOCs of the studied samples, they were measured using the E-nose as a support to the tasting panel. Principal Component Analysis (PCA) was done to reduce the sensors data to two or three principal components. Thus, these components allowed the graphical representation of the grouping of data with similar values. The exploratory PCA (Figure 3) was able to differentiate between olives which black colour fixed by the different procedures (T1–T3). The PC1 explains 74.0% of the total variance of the data, while PC2 explains 17.0%. The E-nose analysis was able to discriminate between olives whose black colour was fixed with ferrous gluconate and olives with fresh and lyophilized Carao. Clearly, when fixing the black colour with fresh and lyophilized Carao, it caused an aromatic profile for the olives that was clearly different. These results are in line with those described in previous sections in the sensory profile and the volatile compounds analysed.

Subsequently, another PCA was carried out with olives in which colour was fixed with fresh and lyophilized Carao stuffed with hydrocolloid flavoured with ‘Mojo picón’ at different concentrations (Figure 4). The E-nose was able to differentiate olives with fresh Carao and ‘Mojo picón’ added. The PC1 explained 94.0% of the total variance of the data, while PC2 explained 3.0% (Figure 4a). The E-nose was also able to discriminate table olives prepared with lyophilized Carao with flavoured hydrocolloid at the different concentrations. The PC1 explained 97.0% of the total variance of the data, while PC2 explained 2.0% (Figure 4b). With this procedure, the negative sensory effects of the addition of Carao during preparation of Californian-style black olives were masked by stuffing olives with flavoured hydrocolloid. The E-nose was used to evaluate its discriminating power as support for the tasting panel. We must highlight that the exploratory PCA was able to differentiate between stuffed olives with different concentrations of ‘Mojo picón’ aroma. Similar results were found when using fresh and lyophilized Carao at different concentrations of the added aromas. Sánchez et al. [27] showed similar results because they separated stuffed black olives with and without added flavouring. This indicates that ‘Mojo picón’ aroma added at a low concentration (2%) is enough to fully achieve our objective of masking the initial aroma of Californian-style black olives prepared with Carao. For all of that, this strategy could be interesting for table olives sector to produce a better-quality product without organoleptic defect.

Subsequently, a classification analysis was performed by PLS-DA using leave-one-out cross-validation. The confusion matrix of the PLS-DA model (Table 4) shows that the sum of the diagonal elements gave a hit rate of 100%. Eight samples from each class were used to construct the model, which allowed to discriminate Californian-style table olives whose colour was fixed with ferrous gluconate and/or fresh and lyophilized Carao. The characteristic aroma of these olives allowed the different classification.

The confusion matrix of the PLS-DA model for fresh and lyophilized Carao with ‘Mojo picón’ aroma (Table 5) shows that the sum of the diagonal elements gave a hit rate of 100% for fresh Carao and 93.8% for lyophilized one. In the confusion matrix, some samples of lyophilized Carao, were wrongly classified between the T6-T7 groups. This may be due to the positive aroma of lyophilized Carao and the low concentration of added aroma. The rest of the samples were successfully predicted with an accuracy of 100%. Therefore, the model predicted a correct discrimination of the samples filled with ‘Mojo picón’ aroma. Sánchez et al. [27] were able to mask the Zapateria defect flavouring stuffed olives improving table olives commercial category.

## 4. Conclusions

The addition of Carao during processing of Californian-style black olives contributed a positive aroma related to the sweet smell, but at the same time they showed negative sensory attributes such as cheese, fermented and metallic flavours/aromas. Furthermore, it was possible to mask the aromas caused by *C. grandis* by stuffing olives with flavoured hydrocolloid with ‘Mojo picón’, and this managed to improve the commercial category of the table olives. The main VOCs detected in fresh and lyophilized Carao with an unpleasant odour were 3-methyl-butanoic acid and 2-methyl-butanoic acid when ‘Mojo picón’ aroma was added to olives, but other positive aromas appeared, such as methyl-ester-2-propenoic acid, dimethyl silanediol, p-cymene and diallyl disulphide. The E-nose discriminated olives based on their aromatic profile, regardless of whether Carao and/or ‘Mojo picón’ had been added during the preparation process. These results agree with the data obtained by the sensory panel and the profile of the VOCs. The E-nose, although it presents some limitations because it must be contrasted and validated with a tasting panel or by studying the profile of VOCs, is a powerful tool that can be used as an alternative to detect chemical compounds in complex samples due to its high discrimination capacity.

## Figures and Tables

**Figure 1 foods-11-02305-f001:**
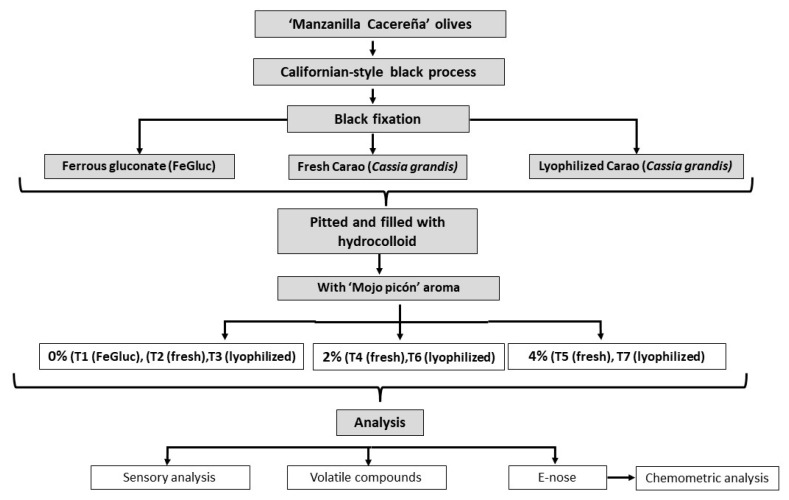
Diagram of the experimental design.

**Figure 2 foods-11-02305-f002:**
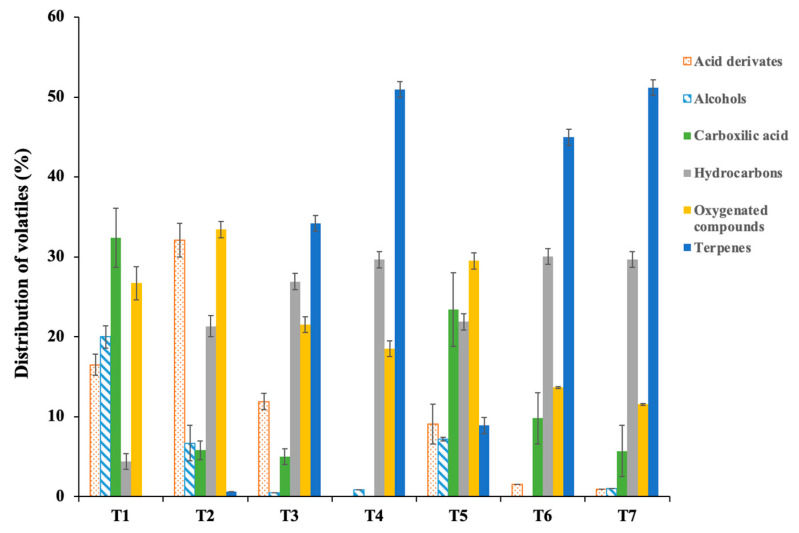
The distribution of the percentage of VOCs in the different treatments. T1: olives with ferrous gluconate stuffed with 0% of ‘Mojo picón’; T2: olives with fresh Carao stuffed with 0% of ‘Mojo picón’; T3: olives with lyophilized Carao stuffed with 0% of ‘Mojo picón’; T4: olives with fresh Carao stuffed with 2% ‘Mojo picón’; T5: olives with fresh Carao stuffed with 4% of ‘Mojo picón’; T6: olives with lyophilized Carao stuffed with 2% of ‘Mojo picón’; T7: olives with lyophilized Carao stuffed with 4% of ‘Mojo picón’.

**Figure 3 foods-11-02305-f003:**
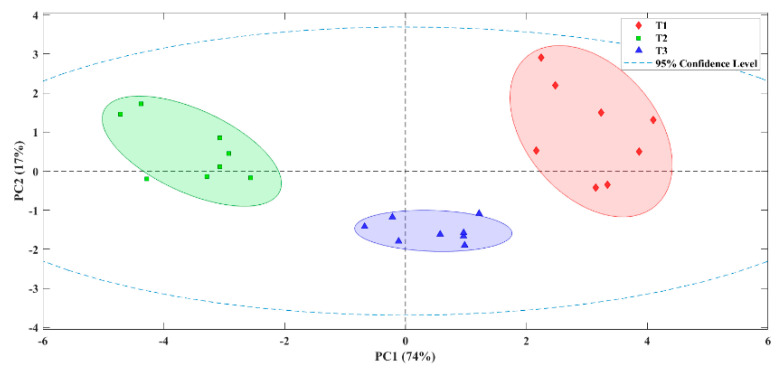
Score plot obtained from the PCA of Californian-style black olives prepared with Carao. T1: olives with ferrous gluconate stuffed with 0% of ‘Mojo picón’; T2: olives with fresh Carao stuffed with 0% of ‘Mojo picón’; T3: olives with lyophilized Carao stuffed with 0% of ‘Mojo picón’.

**Figure 4 foods-11-02305-f004:**
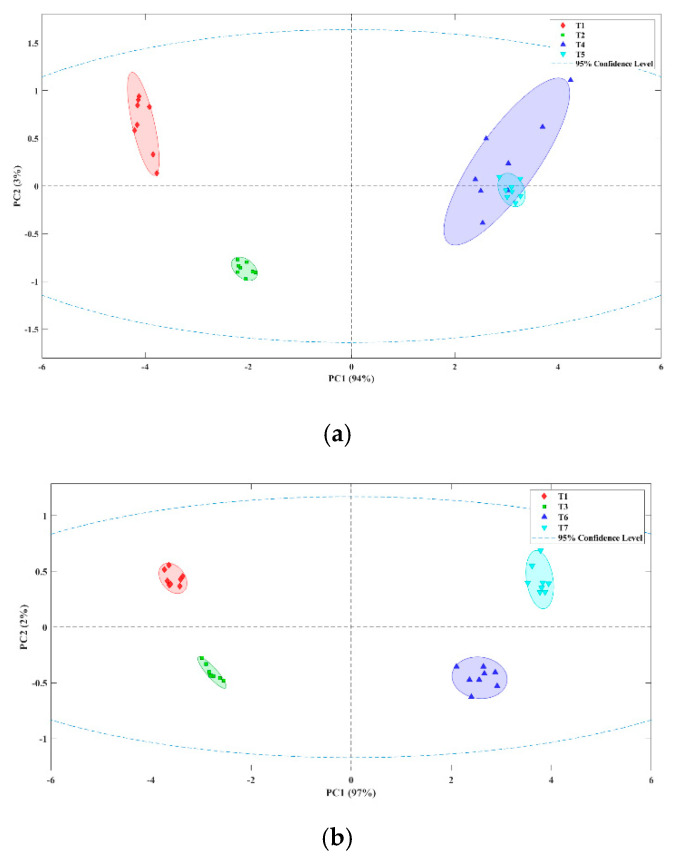
Score plot obtained from the PCA of Californian-style black olives prepared with fresh (**a**) and lyophilized (**b**) Carao and stuffed with flavoured hydrocolloids. T1: olives with ferrous gluconate stuffed with 0% of ‘Mojo picón’; T2: olives with fresh Carao stuffed with 0% of ‘Mojo picón’; T4: olives with fresh Carao stuffed with 2% ‘Mojo picón’; T5: olives with fresh Carao stuffed with 4% of ‘Mojo picón’; T3: olives with lyophilized Carao stuffed with 0% of ‘Mojo picón’; T6: olives with lyophilized Carao stuffed with 2% of ‘Mojo picón’; T7: olives with lyophilized Carao stuffed with 4% of ‘Mojo picón’.

**Table 1 foods-11-02305-t001:** Sensors used in the E-nose.

Sensors	Parameters	Units
Bosch BME680	Temperature	°C
	Pressure	hPa
	Humidity	% RH
	Resistance	W
Sensirion SGP30	eCO_2_	ppm
	TVOC	ppb
	H_2_	
	Ethanol	
ScioSence CCS811	eCO_2_	ppm
	TVOC	ppb
	Resistance	W

**Table 2 foods-11-02305-t002:** Sensory olfactory evaluation of Californian-style black olives.

Samples	Positive Attributes				Negative Attributes				
	Fruity	Sweet	Toasted		Cheesy	Fermented	Metallic	Cooking Effect	Commercial Category
T1	3.0 ± 0.7	n.d.	n.d.	n.d.	n.d.	n.d.	n.d.	2.5 ± 0.7	Extra
T2	4.0 ± 0.8	4.0 ± 0.6	n.d.	n.d.	5.0 ± 0.3	6.0 ± 0.5	5.0 ± 0.4	n.d.	2nd. Category
T4	2.0 ± 0.8	1.5 ± 0.2	n.d.	4.5 ± 0.6	n.d.	n.d.	3.0 ± 0.2	n.d.	1st. Category
T5	1.8 ± 0.3	1.4 ± 0.4	n.d.	6.0 ± 0.5	n.d.	n.d.	n.d.	n.d.	Extra
T3	6.0 ± 0.9	6.0 ± 0.3	2.0 ± 0.4	n.d.	1.5 ± 0.2	1.0 ± 0.3	n.d.	n.d.	Extra
T6	2.5 ± 0.3	3.0 ± 0.2	n.d.	3.0 ± 0.5	n.d.	n.d.	n.d.	n.d.	Extra
T7	n.d.	n.d.	n.d.	5.0 ± 0.4	n.d.	n.d.	n.d.	n.d.	Extra

n.d., not detected. T1: olives with ferrous gluconate stuffed with 0% of ‘Mojo picón’; T2: olives with fresh Carao stuffed with 0% of ‘Mojo picón’; T3: olives with lyophilized Carao stuffed with 0% of ‘Mojo picón’; T4: olives with fresh Carao stuffed with 2% ‘Mojo picón’; T5: olives with fresh Carao stuffed with 4% of ‘Mojo picón’; T6: olives with lyophilized Carao stuffed with 2% of ‘Mojo picón’; T7: olives with lyophilized Carao stuffed with 4% of ‘Mojo picón’.

**Table 3 foods-11-02305-t003:** Profile of volatile compounds for the different treatments applied to Californian-style black olives.

Volatile Compounds	RT (min)	T1	T2	T3	T4	T5	T6	T7
**Acid derivates**								
3-methyl-butanoic acid	10.7	0.0	17.7	8.1	0.0	0.0	0.0	0.0
2-methyl-butanoic acid	11.3	0.0	8.8	3.8	0.0	0.0	0.0	0.0
Ethyl cyclohexanecarboxylate	24.1	16.5	5.6	0.0	0.0	6.4	0.0	0.0
methyl ester-2-Propenoic acid	35.4	0.0	0.0	0.0	1.5	2.7	1.5	0.9
**Alcohol**								
dimethyl-silanediol	4.3	0.0	3.8	0.5	0.9	2.2	0.7	1.0
2-Methoxy-4-methylphenol	27.2	20.0	2.9	0.0	0.0	5.0	0.0	0.0
**Carboxylic acid**								
Benzoic acid	57.6	7.4	2.0	1.5	0.0	19.2	7.3	5.7
Creosol	26.7	25.0	3.8	3.5	0.0	4.2	2.5	0.0
Benzoic acid	0.0	0.0	0.0	0.0	0.0	0.0	0.0	0.0
**Hydrocarbons**								
2,4-dimethyl-hexane	6.5	0.0	2.4	0.3	0.2	7.6	0.5	0.0
Styrene	11.3	4.4	2.0	0.0	0.0	1.7	0.0	0.0
Diallyl disulphide	20.9	0.0	0.0	12.8	11.7	0.0	18.1	17.1
3,7-dimethyl-1,6-Octadien	22.0	0.0	11.9	4.5	3.9	0.0	0.0	0.0
(E)-3-Tetradecene	27.0	0.0	2.7	0.0	0.0	5.3	0.0	0.0
1,2,3,4-tetrahydro-1,1,6-tr-naphthalene	27.3	0.0	1.4	0.0	0.0	3.2	0.0	0.0
dodecamethyl-cyclohexasiloxane	31.4	0.0	0.9	0.0	0.0	4.1	0.0	0.0
di-2-propenyl-trisulfide	31.6	0.0	0.0	7.3	9.2	0.0	8.3	9.1
Allyl trisulfide	31.8	0.0	0.0	2.0	3.1	0.0	2.5	3.5
**Oxygenated compounds**								
Hexanal	6.6	0.0	0.0	1.9	0.7	0.0	0.0	0.0
Benzaldehyde	14.7	23.6	8.7	2.0	1.5	0.0	0.0	0.0
Octanal	16.9	0.0	4.8	1.6	1.0	6.5	0.5	0.1
(Z) 3,7-dimethyl--2,6-octadienal.	19.4	0.0	3.0	0.0	0.0	0.0	0.0	0.0
Bicyclo[3.1.1]heptane-2-carboxaldehyde,	19.9	0.0	1.5	0.0	0.0	0.0	0.0	0.0
(E)-3,7-dimethyl-2,6-Octadienal	20.6	0.0	2.9	0.0	0.0	5.3	0.0	0.0
Nonanal	22.2	0.0	4.0	2.6	0.8	5.2	0.5	0.1
4-(1-methylethyl)-benzaldehyde	29.0	0.0	0.0	6.6	8.1	0.0	7.7	7.0
Cuminaldehyde	29.1	0.0	0.0	2.5	3.4	0.0	2.3	3.6
(E)-2-Decenal	29.9	3.1	8.5	4.4	3.0	12.5	2.7	0.7
**Terpenes**								
β-phellandrene	15.3	0.0	0.0	8.5	8.9	0.0	7.6	9.5
β-pinene	16.1	0.0	0.0	5.5	6.4	0.0	4.7	5.9
p-Cymene	17.9	0.0	0.0	8.0	15.1	4.0	13.8	15.6
D-Limonene	18.1	0.0	0.0	0.0	0.0	0.0	0.0	0.0
β-Terpinene	19.7	0.0	0.0	7.3	14.1	4.9	12.3	13.4
L-α-Terpineol	21.4	0.0	0.0	0.3	0.4	0.0	0.4	0.3
L-β-Terpineol	26.7	0.0	0.6	0.0	0.0	0.0	0.0	0.0
terpinen-7-al	31.0	0.0	0.0	4.6	6.1	0.0	6.2	6.5

R.T.: retention time; T1: olives with ferrous gluconate stuffed with 0% of ‘Mojo picón’; T2: olives with fresh Carao stuffed with 0% of ‘Mojo picón’; T3: olives with lyophilized Carao stuffed with 0% of ‘Mojo picón’; T4: olives with fresh Carao stuffed with 2% ‘Mojo picón’; T5: olives with fresh Carao stuffed with 4% of ‘Mojo picón’; T6: olives with lyophilized Carao stuffed with 2% of ‘Mojo picón’; T7: olives with lyophilized Carao stuffed with 4% of ‘Mojo picón’.

**Table 4 foods-11-02305-t004:** Confusion matrix obtained through PLS-DA for discrimination between stuffed olives with flavoured hydrocolloids. Values are expressed in percentage.

Predicted Class			
Real Class	T1	T2	T3
**T1**	33.3	0	0
**T2**	0	33.3	0
**T3**	0	0	33.3

Diagonal bold contains the percentage of correct assignments. T1: olives with ferrous gluconate stuffed with 0% of ‘Mojo picón’; T2: olives with fresh Carao stuffed with 0% of ‘Mojo picón’; T3: olives with lyophilized Carao stuffed with 0% of ‘Mojo picón’.

**Table 5 foods-11-02305-t005:** Confusion matrix obtained through PLS-DA for discrimination between Californian-style black olives prepared with fresh (a) and lyophilized (b) Carao and stuffed with flavoured hydrocolloids.

**Predicted Class (a)**				
**Real Class**	**T1**	**T2**	**T4**	**T5**
**T1**	25	0	0	0
**T2**	0	25	0	0
**T4**	0	0	25	0
**T5**	0	0	0	25
**Predicted Class (b)**				
**Real Class**	**T1**	**T3**	**T6**	**T7**
**T1**	25	0	0	0
**T3**	0	25	0	0
**T6**	0	0	21.9	3.1
**T7**	0	0	3.1	21.9

Confusion matrix (a) of California-style black olives prepared with fresh and freeze-dried carao (b) and stuffed with flavored hydrocolloids. Diagonal bold contains the percentage of correct assignments. T1: olives with ferrous gluconate stuffed with 0% of ‘Mojo picón’; T2: olives with fresh Carao stuffed with 0% of ‘Mojo picón’; T4: olives with fresh Carao stuffed with 2% ‘Mojo picón’; T5: olives with fresh Carao stuffed with 4% of ‘Mojo picón’; T3: olives with lyophilized Carao stuffed with 0% of ‘Mojo picón’; T6: olives with lyophilized Carao stuffed with 2% of ‘Mojo picón’; T7: olives with lyophilized Carao stuffed with 4% of ‘Mojo picón’.

## Data Availability

The date are available from the corresponding author.

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
