# Peer review of "Masking Effect of Cassia grandis Sensory Defect with Flavoured Stuffed Olives"

_foods, 2022, doi:10.3390/foods11152305_

Round 1
Reviewer 1 Report
It is better to present the sensors used in the e-nose in a tabular format with their characteristics, morphology, materials used, working temperature and detection range.
Please name the gas chromatography instrument used (producer and country) and the detailed process of VOCs extraction.
Figure 4 must be corrected and replaced with a new and improved one. There is error in circling the PCs.
Please highlight the major limitations of the study carried out.
The authors are advised to cite the following articles to improve the manuscript.
- https://doi.org/10.1016/j.foodres.2018.12.057
- https://doi.org/10.3390/proceedings2131061
- https://doi.org/10.3390/foods9040514
The English language and grammatical errors must be corrected/revised throughout the manuscript.
Author Response
Reviewer 1
Comments to Author
It is better to present the sensors used in the e-nose in a tabular format with their characteristics, morphology, materials used, working temperature and detection range.
Thanks for your appreciation. The parameters of the E-nose were placed in table form.
Please name the gas chromatography instrument used (producer and country) and the detailed process of VOCs extraction.
These data have been included in the methodology.
Figure 4 must be corrected and replaced with a new and improved one. There is error in circling the PCs.
Thank you for your comment. The figures 3 and 4 (a and b) have been drawn again with the Matlab program version R2016b, version 9.1.
Highlight the main limitations of the study carried out.
We have highlighted certain limitation of the E-nose through the manuscript.
The authors are advised to cite the following articles to improve the manuscript.
- https://doi.org/10.1016/j.foodres.2018.12.057
- https://doi.org/10.3390/proceedings2131061
- https://doi.org/10.3390/foods9040514
These references have been included in the manuscript.
The English language and grammatical errors must be corrected/revised throughout the manuscript.
Thank you for your comment. Some grammatical and spelling errors have been checked.

Reviewer 2 Report
This work was to study the effect of masking the aroma caused by the addition of Cassia grandis in Californian-style black olives with stuffing olives with flavored hydrocolloid to ‘Mojo picón’ at different concentration.The manuscript is clearly written.
1 There is no space between the number and the ℃,such as line 61, 103 ..., please check. 2 There is some errors in the abbreviation of the reference, please check, like line 500.Author Response
Reviewer 2
Comments to Author
This work was to study the effect of masking the aroma caused by the addition of Cassia grandis in Californian-style black olives with stuffing olives with flavored hydrocolloid to ‘Mojo picón’ at different concentration. The manuscript is clearly written.
Thanks for your appreciation. We have reviewed the manuscript point by point according to the reviewer´s comment.
There is no space between the number and the ℃,such as line 61, 103 ..., please check.
Spaces were placed between the number and ºC throughout the manuscript.
The suggestion has been taken into consideration. Thank you.
There is some errors in the abbreviation of the reference, please check, like line 500.
Thank you. This fact has been modified.

Reviewer 3 Report
the title fully centers the topic of the manuscript.
the manuscript seems to me to be written in a correct and complete form,
except for the introduction, which seemed to me to be a bit poor in citations and could be expanded with more detailed studies concerning the same kind of study that the author did.
Author Response
Reviewer 3
Comments to Author
The title fully centers the topic of the manuscript.
Thanks for your appreciation. We have reviewed the manuscript point by point according to the reviewer.
The manuscript seems to me to be written in a correct and complete form, except for the introduction, which seemed to me to be a bit poor in citations and could be expanded with more detailed studies concerning the same kind of study that the author did.
Thank you for your observations. New references have been included in the introduction part such as:
- https://doi.org/10.1016/j.foodres.2018.12.057
- https://doi.org/10.3390/proceedings2131061
- https://doi.org/10.3390/foods9040514
